# The Real-World Burden of Moderate-to-Severe Psoriasis in Patients Under Systemic Treatment from Baltic Countries: Data from the CRYSTAL Observational Study

**DOI:** 10.3390/medicina61030397

**Published:** 2025-02-25

**Authors:** Maigi Eisen, Ilona Hartmane, Külli Kingo, Ingmars Mikazans, Tiina Toomson, Karin Toomela, Skaidra Valiukeviciene

**Affiliations:** 1Center of Dermatology and Venereology, The North Estonia Medical Centre, 13419 Tallinn, Estonia; 2Department of Dermatology and Venereology, Faculty of Medicine, Riga Stradins University, LV-1010 Riga, Latvia; ilona.hartmane@rsu.lv (I.H.); ingmars.mikazans@rsu.lv (I.M.); 3Department of Dermatology and Venereology, University of Tartu, 50090 Tartu, Estonia; kylli.kingo@kliinikum.ee; 4Dermatology Clinic, Tartu University Hospital, 50417 Tartu, Estonia; 5Department of Dermatovenerology, Pärnu Central Hospital, 80010 Pärnu, Estonia; tiina.toomson@ph.ee; 6AbbVie, 10145 Tallinn, Estonia; 7Department of Skin and Venereal Diseases, Medical Academy, Lithuanian University of Health Sciences (LUHS), LT-44307 Kaunas, Lithuania; skaidra.valiukeviciene@kaunoklinikos.lt; 8Department of Skin and Venereal Diseases, Hospital of LUHS Kauno Klinikos, LT-50161 Kaunas, Lithuania

**Keywords:** drug therapy, patient-reported outcome measures, psoriasis, quality of life, severity of illness index

## Abstract

*Background and Objectives*: Data on disease control, treatment, and quality of life (QoL) in patients with psoriasis from Baltic countries are lacking. In this study, we aimed to assess the disease control, treatment, and QoL of patients with psoriasis in countries from Central and Eastern Europe, and we report data for the Baltic countries. *Materials and Methods*: In a cross-sectional, international study (CRYSTAL), we retrospectively assessed the real-world disease severity and QoL in adult patients (18–75 years) from Estonia, Latvia, and Lithuania with moderate-to-severe psoriasis receiving continuous systemic treatment ≥ 24 weeks. Analyses included 50 patients from each country and were descriptive. *Results*: The median disease duration was 15.2–19.9 years across the countries. Most patients (78.0% in Estonia, 100% in Latvia, and 68.0% in Lithuania) were receiving monotherapy with biological agents, mainly TNF inhibitors. An absolute PASI score ≤ 3 was achieved by 82.0%, 70.0%, and 64.0% of patients in the overall study population and 89.7%, 70.0%, and 61.8% of patients receiving biologic monotherapy in Estonia, Latvia, and Lithuania, respectively. Across the countries, impairments in QoL as expressed by a Dermatology Life Quality Index score > 5 were reported by 14.0–34.0% of patients, while 88.0–96.0% of patients were satisfied with their treatment. *Conclusions*: Although most patients showed low absolute PASI scores and satisfaction with their evolution after ≥24 weeks of systemic treatment, they still reported an impact on QoL. This finding underlines that further optimization of systemic treatment strategies is needed to improve outcomes in moderate-to-severe psoriasis in Baltic countries.

## 1. Introduction

Psoriasis, a chronic, immune-mediated inflammatory disease of the skin, is estimated to affect up to 4% of the general population, with a higher prevalence reported in high-income countries and regions [1,2]. The incidence of psoriasis has shown an increasing trend over time since the 1990s [3], and in 2019, 4.6 million incident cases were estimated to occur globally [4]. In addition, psoriasis impacts patients’ quality of life (QoL) to a large extent [5,6], generating a considerable economic burden for both the patient and society [7,8].

Epidemiologic information from most countries and regions is still lacking to date, even for high-income settings. In the three Baltic countries, psoriasis is estimated to have a low prevalence, but higher in the adult than the pediatric population [1,9], although data are limited. In 2021, Hartmane et al. reported that there were around 35,000 psoriasis patients in Estonia, 40,000 in Latvia, and 76,000 in Lithuania, and that prevalence rates were similar and relatively low (≤2%) in all three countries [10].

Psoriasis severity and the efficacy of available treatments are evaluated using the Psoriasis Area and Severity Index (PASI) [11]. A ≥75% reduction from the baseline (PASI 75) in PASI scores after treatment has been the therapeutic goal for a long time. However, with the advent of biological therapy, PASI 90 or even complete skin clearance (PASI 100) after six months of treatment has become a very realistic target to achieve for moderate-to-severe psoriasis patients [12,13]. Despite the difference in skin clearance, a previous study showed that the QoL of patients achieving PASI 75 versus PASI 90 might not be significantly different from a clinical perspective [14]. It is now generally agreed that an absolute PASI score (not relative to baseline PASI) of ≤2 or ≤3 is a better therapeutic target and also correlates better with Dermatology Life Quality Index (DLQI) scores than relative PASI [15,16].

The recommendations of EuroGuiDerm include the use of systemic therapy for patients with moderate-to-severe psoriasis with either conventional or biological agents [17]. Biologics have been shown to induce adequate clinical response rates and acceptability by patients in real-world settings [18,19], both of which are improved as compared to conventional agents [20,21]. However, their use is still not uniformly implemented in Central and Eastern Europe (CEE) [22], and treatment initiation with biologics can differ considerably across European countries, as several countries adhere to national guidelines or have different reimbursement rules. In addition, there is little available published data on the type of treatments utilized in routine practice [22,23]. To address this knowledge gap, we conducted an observational study (CRYSTAL study) aiming at describing psoriasis severity through absolute PASI scores in patients with moderate-to-severe psoriasis receiving systemic treatment for ≥24 weeks in routine clinical practice in several CEE countries. The study further describes treatment patterns, QoL, impairment of work and activity, as well as patients’ treatment satisfaction [24]. In the Baltic states, there are incipient or no public registries to collect data on psoriasis severity, patient characteristics, treatment, and QoL in a comprehensive, standardized mode. Here, we report the results of the CRYSTAL study for Estonia, Latvia, and Lithuania.

## 2. Materials and Methods

### 2.1. Study Design and Participants

The CRYSTAL study was a multi-country, multicenter, cross-sectional/retrospective study performed between September 2020 and February 2021 in seven CEE countries. Participants from Estonia, Latvia, and Lithuania were enrolled in 1 private and 5 public (of which 4 were university hospitals) hospital-based centers/clinics specialized in dermatology. All participants were assessed for their eligibility and enrolled in a single visit, when the informed consent form was signed and study data were extracted. Treatments for psoriasis were prescribed according to routine practice in each country. National and local ethics committees approved the study in all participating countries. The CRYSTAL study was designed and conducted according to the principles stated in the Declaration of Helsinki, the Good Pharmacoepidemiology Practices (GPP) guidelines of the International Society for Pharmacoepidemiology (ISPE), and the regulations of each country. The CRYSTAL study was publicly registered in the former European Union electronic Register of Post-Authorization Studies (EU PAS Register^®^) under the EU PAS number EUPAS36459.

Criteria for patients’ inclusion in the study were as follows: age between 18 and 75 years, an established diagnosis of moderate-to-severe chronic plaque psoriasis, and use of any systemic treatment approved for psoriasis (either monotherapy or a combination treatment) administered on a continuous basis for ≥24 weeks. Eligible patients were required to have their absolute PASI score evaluated at the initiation of their current systemic therapy (with a window of 30 days prior and 7 days after allowed per protocol); another assessment was planned at the time of enrollment (i.e., the study visit). Administration of any investigational treatment within one month or five half-lives of the agent led to exclusion from the study.

The objectives of the CRYSTAL study were previously reported in detail [24]. Here, we present an analysis by country for the three Baltic countries (Estonia, Latvia, and Lithuania) on the characteristics of the disease, treatment, QoL, and satisfaction with treatment outcomes of patients with moderate-to-severe psoriasis routinely managed with systemic therapy.

### 2.2. Data Collection

At the study visit, the data collected included current sociodemographic, anthropometric, and lifestyle characteristics; comorbidities; disease characteristics and severity by absolute PASI; and treatment for psoriasis. Patient-reported outcomes in terms of QoL, work productivity and activity impairment (WPAI), and patient satisfaction with treatment were also collected. Retrospective data comprised disease characteristics at the time of psoriasis diagnosis, relevant medical history, prior psoriasis treatments, and the status of current treatment from its start until enrollment day (i.e., start date, starting dose, and dose escalations). A password-protected, web-based data capture (WBDC) electronic system was used for data collection by the study investigator at each site.

The scores of the questionnaires completed by patients, namely the DLQI, EQ-5D-5L (including the EuroQol Visual Analogue Scale (EQ-VAS)), WPAI-PSO, and patient satisfaction questionnaire (paper format), were calculated as indicated in Appendix A.

### 2.3. Statistical Analysis

The sample size calculation for the entire study was previously described in detail. Sample size considerations were not applied by country, but a minimum of 50 patients from each country was planned for enrollment.

Statistical analyses were descriptive; analyses were also performed by country for selected objectives (a list of endpoints is presented in Appendix A). All analyses were performed on the full analysis set (FAS), which included all eligible patients from each country with available data. Additional analyses (as feasible) were conducted in subgroups stratified by current systemic therapy and by the absolute PASI score as assessed at the study visit.

SAS statistical software package was used.

## 3. Results

### 3.1. Patient, Disease, and Treatment Characteristics

In each country, 50 patients were enrolled, and all were included in the full analysis set. Participants were enrolled at public (for 140 patients, of whom 120 attended university hospitals) and private (for 10 patients) hospitals/clinics.

The median age at enrollment was 45.5 years in Estonia, 47.0 years in Latvia, and 49.8 years in Lithuania. Most patients (≥68.0% in each country) were male. Most patients (48.0% in Estonia, 76.0% in Latvia, and 54.0% in Lithuania) had ongoing comorbidities at study visit (Table 1).

The proportion of patients with active psoriatic arthritis varied across the countries: 4.0% in Estonia, 16.0% in Latvia, and 24.0% in Lithuania. The median age at the onset of psoriasis signs and symptoms was 20.7 years in Estonia and Latvia and 17.9 years in Lithuania. The median disease duration ranged across countries from 15.2 years (Lithuania) to 19.9 years (Latvia) (Table 2).

At the current systemic treatment initiation, almost all patients (100% in Estonia and Latvia and 94.0% in Lithuania) had received at least one other prior therapy which had been discontinued. Across the countries, 28.0–54.0% and 48.0–98.0% of patients had received a prior biologic and non-biologic systemic treatment, respectively. In addition, 68.0–100% of patients had received at least one prior topical treatment, and 70.0–84.0% had received prior photo (chemo) therapy. The prior systemic treatment was with biologic agents for 54.0% of patients in Estonia, 50.0% in Latvia, and 28.0% in Lithuania.

The median period of receiving the current systemic treatment was 28.2 months in Estonia, 18.4 months in Latvia, and 28.9 months in Lithuania. Most patients (92.0% in Estonia, all in Latvia, and 72.0% in Lithuania) were receiving monotherapy; biological agents (mainly tumor necrosis factor (TNF) inhibitors) were used for 78.0%, 100%, and 68.0% of patients in Estonia, Latvia, and Lithuania, respectively. Combination therapy was used for 8.0% of patients in Estonia and 28.0% of those in Lithuania (Table 3). Among patients receiving biological monotherapy, treatment was escalated (i.e., either the dose was increased or the time between doses was decreased) in at least 2 (5.1%) patients in Estonia, 1 (2.0%) patient in Latvia, and 7 (20.6%) patients in Lithuania, mostly due to insufficient response.

### 3.2. PASI Score

The mean (standard deviation (SD)) absolute PASI score at the study visit was 1.6 (2.3) in Estonia, 4.0 (6.2) in Latvia, and 3.7 (4.8) in Lithuania. The scores varied across the countries and systemic treatments received, with a trend for lower PASI scores observed in Estonia and for combination therapy (Table 3). The proportion of patients with an absolute PASI score ≤ 3 was 82.0%, 70.0%, and 64.0% in the overall study population and 89.7%, 70.0%, and 61.8% in the population treated with monotherapy with biological agents in Estonia, Latvia, and Lithuania, respectively (Figure 1). An absolute PASI score ≤ 1 was achieved by 66.7% (Estonia), 48.0% (Latvia), and 26.5% (Lithuania) of patients receiving monotherapy with biologics (Appendix A).

### 3.3. Patient-Reported Outcomes

The median DLQI total score in the overall study population ranged from 0.5 to 2.0 across the three countries (Table 4). Impairments in QoL, as indicated by DLQI scores <2–≤5 and >5, were noticed in 20.0% and 14.0% of patients in Estonia, 20.0% and 18.0% of patients in Latvia, and 18.0% and 34.0% of patients in Lithuania (Figure 2).

All patients completed the QoL questionnaires. The median EQ-5D-5L utility score in the overall population ranged from 0.8 to 1.0 across the three countries. The most frequently reported negatively affected dimension was pain/discomfort (by 44.0%, 28.0%, and 62.0% of patients in Estonia, Latvia, and Lithuania, respectively), followed by anxiety/depression and mobility. The median EQ-VAS score was 80.0 in all countries (Table 4).

Of the patients who completed the WPAI-PSO questionnaire, most were employed (82.0% in Estonia and Latvia and 72.0% in Lithuania) (Table 4). The highest impact of disease severity was reported in the domains of activity impairment and work productivity loss. At higher PASI scores, a trend for higher impacts was noticed for all domains (Appendix A).

All patients completed the patient satisfaction questionnaires. In the overall population from each country, higher percentages of patients (96.0% in Estonia, 88.0% in Latvia, and 88.0% in Lithuania) reported satisfaction with the overall control of their disease obtained under their current systemic therapy (Table 4).

## 4. Discussion

We provide recent and reliable evidence on the characteristics, treatment patterns, disease control, and QoL of psoriasis patients with moderate-to-severe disease from Estonia, Latvia, and Lithuania, managed with systemic therapy in routine clinical settings.

In all three countries, the study population was characterized by a relatively young age at the onset of psoriasis signs and symptoms (17.9–20.7 years), lower than that observed in the entire population of the CRYSTAL study (26.0 years). Psoriasis was diagnosed in less than 1 year from the onset of signs and symptoms in patients from Estonia and Latvia. In contrast, the median time to diagnosis in Lithuania was approximately 3.1 years. Patients were predominantly male (≥68.0%), similar to observations in the entire CRYSTAL study population [24] and a previous study from 2008 including 913 patients with psoriasis from CEE [25].

We found that most patients in the Baltic countries (≥68%) were treated with biological monotherapy, mainly a TNF inhibitor agent. This was especially true in Latvia, where all patients were receiving monotherapy with biologics, mostly with adalimumab, followed by ustekinumab. In Estonia, most patients were receiving monotherapy with a TNF inhibitor, with adalimumab being used more frequently. A small proportion of patients were undergoing monotherapy with methotrexate or ciclosporin (14.0%) or combination therapy (8.0%) using mainly methotrexate and a TNF inhibitor (infliximab or adalimumab). In Lithuania, a larger proportion of patients (28.0%) received treatment with combination therapy, including methotrexate and mostly a TNF inhibitor (infliximab or etanercept).

This difference in terms of the systemic treatment received reflects the current therapeutic landscape in the Baltic countries, as well as the different criteria for reimbursement or access to specific agents. The most recent recommendations in Europe (the EuroGuiDerm guideline) state that systemic treatment for moderate-to-severe cases of psoriasis should be initiated with conventional agents as the first line (1L) and biological agents in the event of an inadequate response, contraindications, or intolerance to the conventional systemic treatment. Still, for severe psoriasis, and when failure of conventional agents is anticipated, biological agents are recommended as a 1L treatment [17]. However, several European countries, such as France or the United Kingdom [26,27], follow national guidelines. Biological therapy began to be reimbursed in 2018 in Latvia and in 2011 in Estonia and Lithuania. In Estonia, treatment with biologics is indicated for a disease duration ≥ 6 months, a PASI or body surface area (BSA) score ≥ 10, and/or a DLQI score ≥ 10 (meaning that treatment can be initiated when the impact on QoL is high, even at lower BSA scores). In addition, therapy with biological agents can only be initiated if an alternative standard systemic treatment (including with acitretin, cyclosporine, methotrexate, narrow-band ultraviolet B, and psoralen + ultraviolet A photochemotherapy) is contraindicated, if the previous treatment needs to be discontinued due to side effects, or if no treatment effect is observed after using at least two different systemic treatments. Biological therapy is initiated for a form of psoriasis that requires repeated hospitalizations or for unstable, life-threatening forms (erythrodermic or pustular psoriasis). Any dermato-venereologist can initiate the treatment [10]. In Latvia, where national guidelines are in place, systemic treatment with biological agents is initiated for patients with moderate and severe chronic psoriasis (PASI and BSA scores ≥ 10), for whom other systemic therapy with cyclosporine or methotrexate and phototherapy has not been effective or is contraindicated/not tolerated. Treatment can be prescribed by a dermato-venereology specialist, following the decision made by a council at a tertiary level of care in an outpatient and/or inpatient dermato-venerology center [10]. In Lithuania, during the first 3 years after the introduction of biological therapy, biological drugs (usually infliximab or etanercept) were purchased centrally by state health insurance funds, in university hospitals, for approximately 100 patients. Since 2017, systemic treatment with biologics has been indicated, typically with a TNF inhibitor as 1L and an IL-12/23 inhibitor (usually ustekinumab), as a second line of treatment. At the time the CRYSTAL trial was conducted, systemic treatment with biologics was initiated for moderate to severe psoriasis patients (PASI and DLQI scores > 10, psoriasis duration ≥ 6 months) when systemic treatment with methotrexate or acitretin for ≥6 months (currently ≥3) was ineffective or poorly tolerated. Only university hospital-based dermatological services can initiate the treatment [10]. In all countries, the 1L biological agent is anti-TNF, with this choice being driven by the lower cost compared with other biologics available. However, since 2022, cost restrictions were lifted in Lithuania for all biologics. The therapeutic goals are relative PASI scores > 75 in Estonia and Lithuania and >90 in Latvia, and patients are evaluated periodically by dermatologists to decide on the need to discontinue or progress to second- or third-line biological therapy [10]. Based on data from health insurance funds until 2020, the numbers of patients with psoriasis receiving biological therapy in the Baltic countries varied, with 214 patients in Estonia, 113 patients in Latvia, and 330 patients in Lithuania. Reported to 100,000 inhabitants, this is equivalent to 6 psoriasis patients in Latvia, 12 in Lithuania, and 16 in Estonia receiving biologics, revealing a 1.3–2.6-fold difference between the countries in biologics use [28,29,30]. Considering the prevalence of psoriasis in each country, these figures indicate suboptimal access to treatment due to multiple factors, including reimbursement conditions, availability of treatments, and long-term or overuse of topical steroids (Latvia and Lithuania) [10]. However, in our analysis including only patients with severe and moderate psoriasis during 2020–2021, we observed considerably higher proportions of patients treated with biologics, comparable to those reported for the entire population of the CRYSTAL study [24].

We also found that most patients (64.0–82.0%) undergoing systemic treatment (mainly with biologics) in all three countries had absolute PASI scores ≤ 3, indicative of treatment success [31]. There was a trend for higher proportions of patients with PASI scores ≤ 3 in patients receiving monotherapy with biological agents versus those receiving non-biologics in Estonia and Lithuania. Our findings are similar to those observed in the entire study population [24] and suggest the better efficacy of systemic biologic treatment compared to conventional therapy [13]. Notably, there was a higher proportion of patients achieving PASI scores ≤ 3 in Estonia compared to the other two countries, while psoriatic arthritis was relatively more common among patients in Latvia (16.0%) and Lithuania (24.0%) than in Estonia (4.0%). Patients with psoriatic arthritis are known to have more extensive skin disease and higher PASI and BSA scores than those without [32].

Across the three Baltic countries, 48.0–62.0% of patients had DLQI scores > 1 and 14.0–34.0% of patients had scores > 5, with the values observed in the entire CRYSTAL population being in these ranges as well [24]. The overall DLQI score was considerably lower than that observed in a previous study conducted in biologically naïve patients from CEE [25]. An impact on the QoL was observed, especially in terms of pain/discomfort, emphasizing the multifaceted consequences of psoriasis that should be further explored. A previous study conducted in Lithuania also showed that psoriasis patients experienced a deterioration in their QoL, depression, and anxiety, irrespective of disease severity [33]. A more recent study in Lithuanian patients showed reduced DLQI scores compared to baseline in all patients receiving biological therapy after nine months of treatment [34]. In this study, we also observed a trend for lower DLQI scores in Estonia compared to Latvia and Lithuania. This may be partly explained by the lower proportion of patients with active psoriatic arthritis (potentially indicating a higher prevalence of extensive skin disease [32]) in Estonia in the current study.

For all three countries, the WPAI-PSO scores were low in this population mostly treated with biological agents, even among patients with PASI scores > 5. The satisfaction of patients with their current systemic therapy was high, with 88.0–96.00% of patients in the three Baltic states being satisfied with their treatment, compared to 90.5% of patients from all CEE countries included in the study [24]. Although these are encouraging results, the disease impact on QoL despite current treatment should be explored to a larger extent at the patient level. In addition, the high proportion of patients with comorbidities across the countries (62.0–78.0%) underlines the importance of comprehensive patient management, aiming to control all associated diseases.

The study limitations include those inherent to the cross-sectional and retrospective design, including patient selection bias and patient recall bias. However, we undertook measures to minimize these by enrolling patients in a consecutive manner at the study sites and by applying simple questionnaires, with a short or no recall period. The study was conducted during the COVID-19 pandemic, which had a presumably negative impact on the patients’ mental health and perceived QoL. The small sample size for each country is an additional limitation. The overall study population differed among the countries in terms of the treatment received, although it was homogenous in terms of sociodemographic characteristics. Due to the descriptive nature of the analyses, differences between countries should be interpreted with caution. To better characterize the disease and treatment responses in more diverse patient populations across the Baltic countries, future studies including larger sample sizes are needed.

## 5. Conclusions

Moderate-to-severe psoriasis patients are treated in Baltic countries predominantly with TNF inhibitors after failure of systemic conventional therapy. Although variations are observed, reflecting different therapeutic guidelines and reimbursement strategies, most patients show low absolute PASI scores and satisfaction with their evolution after at least 24 weeks of continuous treatment. Nevertheless, a negative impact on QoL was still reported, underscoring the need to further optimize the therapeutic strategies for greater disease control.

## Figures and Tables

**Figure 1 medicina-61-00397-f001:**
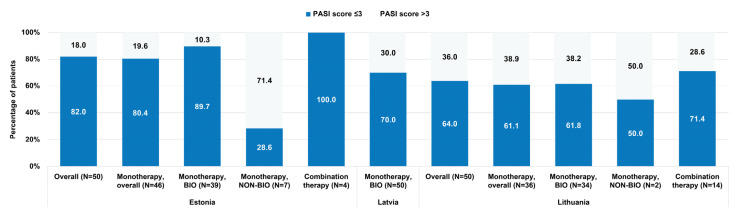
Distribution of patients by Psoriasis Area and Severity Index (PASI) scores ≤ 3 and >3 at study visit and systemic treatment (Monotherapy, BIO (patients treated with biological agents); Monotherapy, NON-BIO (patients treated with conventional agents), and Combination therapy) in each country (full analysis set).

**Figure 2 medicina-61-00397-f002:**
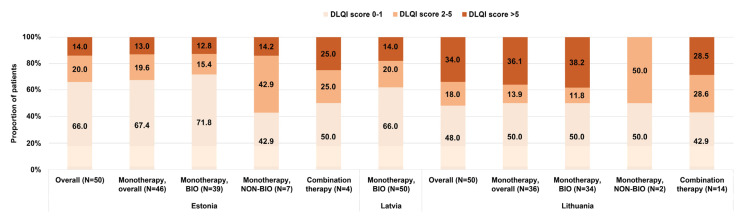
Distribution of patients by Dermatology Life Quality Index (DLQI) score in the overall population and by current systemic therapy (Monotherapy, BIO (patients treated with biological agents); Monotherapy, NON-BIO (patients treated with conventional agents), and Combination therapy) in each country (full analysis sets).

**Table 1 medicina-61-00397-t001:** Patient characteristics at the study visit (full analysis set in each country).

Characteristic	Estonia (N = 50)	Latvia (N = 50)	Lithuania (N = 50)
**Sociodemographics**			
Median (IQR) age, years	45.5 (39.6–55.7)	47.0 (37.1–56.6)	49.8 (35.1–59.8)
Male sex, n (%)	37 (74.0%)	36 (72.0%)	34 (68.0%)
Employment status, n (%)			
Employed (paid employee/self-employed)	41 (82.0%)	41 (82.0%)	35 (70.0%)
Unemployed	7 (14.0%)	4 (8.0%)	5 (10.0%)
Retired	1 (2.0%)	3 (6.0%)	8 (16.0%)
Student	1 (2.0%)	2 (4.0%)	2 (4.0%)
**Anthropometrics**			
Mean (SD) weight, kg	93.9 (19.1)	90.3 (18.0)	88.7 (21.0)
Mean (SD) height, cm	178.0 (8.7)	177.0 (9.4)	174.0 (10.1)
Mean (SD) body mass index, kg/m^2^	29.7 (5.9)	28.7 (4.7)	29.3 (6.7)
**Lifestyle**			
Smoking status, n (%)			
Never a smoker	12 (24.0%)	26 (52.0%)	21 (42.0%)
Occasional smoker	3 (6.0%)	5 (10.0%)	0 (0.0%)
Current smoker	18 (36.0%)	9 (18.0%)	10 (20.0%)
Former smoker	17 (34.0%)	10 (20.0%)	19 (38.0%)
Alcohol consumption in the last month, n (%)			
No alcohol consumption	6 (12.0%)	17 (34.0%)	24 (48.0%)
Occasional (1–2 units/week)	39 (78.0%)	32 (64.0%)	23 (46.0%)
Regular (>2 units/week)	4 (8.0%)	1 (2.0%)	3 (6.0%)
Not reported/unknown	1 (2.0%)	0 (0.0%)	0 (0.0%)
**Comorbidities**			
Median (IQR) number of comorbidities at study visit	0.0 (0.0–2.0)	1.0 (1.0–3.0)	1.0 (0.0–1.0)
Clinically significant medical/surgical history and comorbidities, n (%)			
Overall	31 (62.0%)	39 (78.0%)	31 (62.0%)
Vascular disorders	14 (28.0%)	14 (28.0%)	18 (36.0%)
Metabolism and nutrition disorders	13 (26.0%)	16 (32.0%)	8 (16.0%)
Cardiac disorders	3 (6.0%)	2 (4.0%)	6 (12.0%)
Surgical and medical procedures	6 (12.0%)	-	6 (12.0%)
Endocrine disorders	1 (2.0%)	15 (30.0%)	4 (8.0%)
Musculoskeletal and connective tissue disorders	7 (14.0%)	4 (8.0%)	3 (6.0%)
Infections and infestations	6 (12.0%)	5 (10.0%)	2 (4.0%)
Hepatobiliary disorders	3 (6.0%)	12 (24.0%)	1 (2.0%)
Gastrointestinal disorders	-	9 (18.0%)	2 (4.0%)
Respiratory, thoracic and mediastinal disorders	2 (4.0%)	7 (14.0%)	3 (6.0%)
Past	18 (36.0%)	11 (22.0%)	12 (24.0%)
Ongoing	24 (48.0%)	38 (76.0%)	27 (54.0%)

IQR: interquartile range; N: total number of patients; n (%): number (percentage) of patients in each category; SD: standard deviation. Note: Comorbidities with an incidence of at least 10% in each country have been added. The comorbidities were classified using the Medical Dictionary for Regulatory Activities (MedDRA) v24.0.

**Table 2 medicina-61-00397-t002:** Disease characteristics (full analysis set in each country).

Characteristic	Estonia (N = 50)	Latvia (N = 50)	Lithuania (N = 50)
Median (IQR) age at onset of psoriasis signs and symptoms, years	20.7 (14.1–31.5)	20.7 (11.9–30.7)	17.9 (15.1–27.4)
Median (IQR) age at plaque psoriasis diagnosis, years	22.9 (13.9–35.3)	21.0 (13.6–33.3)	29.4 (17.7–40.9)
Median (IQR) time from the onset of psoriasis signs and symptoms to diagnosis of plaque psoriasis, years	0.0 (0.0–0.9)	0.0 (0.0–1.0)	3.1 (0.0–15.8)
Median (IQR) time from diagnosis of plaque psoriasis to study visit, years	19.0 (13.0–34.9)	19.9 (11.7–27.7)	15.2 (4.5–27.0)
Presence of psoriatic plaques, n (%)	31 (62.0%)	36 (72.0%)	48 (96.0%)
Psoriasis severity at initial diagnosis, n (%)			
Mild	15 (30.0%)	13 (26.0%)	19 (38.0%)
Moderate	13 (26.0%)	9 (18.0%)	8 (16.0%)
Severe	3 (6.0%)	0 (0.0%)	22 (44.0%)
Positive family history of psoriasis, n (%)	21 (42.0%)	29 (58.0%)	27 (54.0%)
History of psoriatic arthritis, n (%)	17 (34.0%)	17 (34.0%)	26 (52.0%)
Active psoriatic arthritis, n (%)	2 (4.0%)	8 (16.0%)	12 (24.0%)
Presence of dactylitis, n (%)	2 (4.0%)	1 (2.0%)	1 (2.0%)
Presence of spondylitis, n (%)	3 (6.0%)	3 (6.0%)	1 (2.0%)
Presence of enthesitis, n (%)	2 (4.0%)	8 (16.0%)	0 (0.0%)
Presence of nail psoriasis, n (%)	10 (20.0%)	9 (18.0%)	25 (50.0%)
Severe itching or pruritus over the last 7 days, n (%)	6 (12.0%)	16 (32.0%)	20 (40.0%)

IQR: interquartile range; N: total number of patients; n (%): number (percentage) of patients in each category.

**Table 3 medicina-61-00397-t003:** Current systemic treatment characteristics (full analysis set in each country).

Characteristic	Estonia (N = 50)	Latvia (N = 50)	Lithuania (N = 50)
**Current systemic treatment for psoriasis, n (%)**			
Monotherapy	46 (92.0%)	50 (100%)	36 (72.0%)
Monotherapy with biological agent	39 (78.0%))	50 (100%)	34 (68.0%)
TNF inhibitor	20 (40.0%)	29 (58.0%)	25 (50.0%)
Adalimumab	12 (24.0%)	29 (58.0%)	14 (28.0%)
Infliximab	7 (14.0%)	0 (0.0%)	2 (4.0%)
Etanercept	1 (2.0%)	0 (0.0%)	9 (18.0%)
IL-17 inhibitor (secukinumab)	7 (14.0%)	1 (2.0%)	0 (0.0%)
IL-12/23 inhibitor (ustekinumab)	10 (20.0%)	20 (40.0%)	9 (18.0%)
IL-23 inhibitor	2 (4.0%)	0 (0.0%)	0 (0.0%)
Guselkumab	1 (2.0%)	0 (0.0%)	0 (0.0%)
Risankizumab	1 (2.0%)	0 (0.0%)	0 (0.0%)
Monotherapy with conventional agent	7 (14.0%)	0 (0.0%)	2 (4.0%)
Methotrexate	6 (12.0%)	0 (0.0%)	2 (4.0%)
Ciclosporin	1 (2.0%)	0 (0.0%)	0 (0.0%)
Combination therapy	4 (8.0%)	0 (0.0%)	14 (28.0%)
TNF inhibitor + conventional agent	3 (6.0%)	0 (0.0%)	12 (34.0%)
Methotrexate + adalimumab	1 (2.0%)	0 (0.0%)	1 (2.0%)
Methotrexate + infliximab	2 (4.0%)	0 (0.0%)	8 (16.0%)
Methotrexate + etanercept	0 (0.0%)	0 (0.0%)	3 (6.0%)
IL-12/23 inhibitor + conventional agent	1 (2.0%)	0 (0.0%)	2 (4.0%)
(ustekinumab + methotrexate)			
IL-17 inhibitor + conventional agent	0 (0.0%)	0 (0.0%)	0 (0.0%)
**Median (IQR) duration of current systemic treatment, months**			
Overall	28.2 (14.9–46.1)	18.4 (13.8–22.4)	28.9 (12.2–57.5)
Monotherapy	26.9 (14.0–46.0)	18.4 (13.8–22.4)	19.4 (11.7–47.6)
Biological agents	28.4 (13.3–47.5)	18.4 (13.8–22.4)	19.4 (11.8–48.6)
Conventional agents	25.8 (14.9–37.3)	-	18.9 (8.8–29.0)
Combination therapy	55.6 (32.7–92.5)	-	51.5 (41.5–92.1)
**Median (IQR) PASI score at the initiation of current systemic treatment**			
Overall	13.7 (10.6–16.0)	18.5 (15.0–24.0)	28.9 (12.2–57.7)
Monotherapy	13.9 (10.6–16.8)	18.5 (15.0–24.0)	19.4 (11.7–47.6)
Biological agents	13.8 (10.6–16.0)	18.5 (15.0–24.0)	19.4 (11.8–48.6)
Conventional agents	15.6 (6.7–25.0)	-	18.9 (8.8–29.0)
Combination therapy	13.2 (10.2–14.8)	-	51.5 (41.5–92.1)
**Mean (SD) PASI score at study visit**			
Overall	1.6 (2.3)	4.0 (6.2)	3.7 (4.8)
Monotherapy	1.6 (2.4)	4.0 (6.2)	4.1 (5.2)
Biological agents	1.1 (1.8)	4.0 (6.2)	3.9 (5.3)
Conventional agents	4.6 (3.4)	-	6.7 (5.3)
Combination therapy	1.1 (0.9)	-	2.9 (3.5)
**Median (IQR) PASI score at study visit**			
Overall	0.7 (0.0–2.8)	1.2 (0.0–5.0)	1.8 (0.9–5.4)
Monotherapy	0.5 (0.0–1.2)	1.2 (0.0–5.0)	2.1 (1.1–5.5)
Biological agents	0.6 (0.0–3.0)	1.2 (0.0–5.0)	1.9 (1.0–5.4)
Conventional agents	3.4 (3.0–5.0)	-	6.7 (2.9–10.4)
Combination therapy	1.2 (0.4–1.9)	-	1.6 (0.4–4.8)

N: total number of patients; n (%): number (percentage) of patients in each category; IL: interleukin; IQR: interquartile range; PASI: Psoriasis Area and Severity Index; SD: standard deviation; TNF: tumor necrosis factor. Note: The duration of treatment (expressed in months) was calculated using the following formula: (Date of study visit—Date of treatment onset + 1)/30.42. Imputation was used for all partial missing dates, as follows: (a) for treatment start dates, when only the day or the month was missing, they were set as the first day of the month or the first month of the year, respectively; (b) for treatment end dates, the reverse was applied; (c) when only the year was available, the day and the month were imputed as previously described in (a) and (b).

**Table 4 medicina-61-00397-t004:** Patient-reported outcomes at study visit (full analysis set in each country).

Characteristic	Estonia (N = 50)	Latvia (N = 50)	Lithuania (N = 50)
**Dermatology-specific health-related quality of life**			
Median DLQI (IQR) score	0.5 (0.0–2.0)	1.0 (0.0–4.0)	2.0 (0.0–8.0)
Symptoms and feelings domain	0.0 (0.0–1.0)	1.0 (0.0–1.0)	1.0 (0.0–2.0)
Daily activity domain	0.0 (0.0–0.0)	0.0 (0.0–1.0)	0.0 (0.0–2.0)
Leisure domain	0.0 (0.0–0.0)	0.0 (0.0–1.0)	0.0 (0.0–1.0)
Work and school domain	0.0 (0.0–0.0)	0.0 (0.0–0.0)	0.0 (0.0–0.0)
Personal relationships domain	0.0 (0.0–0.0)	0.0 (0.0–0.0)	0.0 (0.0–1.0)
Treatment domain	0.0 (0.0–1.0)	0.0 (0.0–0.0)	0.0 (0.0–1.0)
**General health-related QoL based on EQ-5D-5L at the study visit**			
**Percentage of patients with problems reported for each EQ-5D-5L dimension, n (%)**			
Mobility	12 (24.0%)	9 (18.0%)	16 (32.0%)
Self-care	7 (14.0%)	7 (14.0%)	14 (28.0%)
Usual activities	10 (20.0%)	11 (22.0%)	15 (30.0%)
Pain/discomfort	22 (44.0%)	14 (28.0%)	31 (62.0%)
Anxiety/depression	20 (40.0%)	14 (28.0%)	17 (34.0%)
Median (IQR) EQ-5D-5L utility index score	0.9 (0.7–1.0)	1.0 (0.8–1.0)	0.8 (0.6–1.0)
Median (IQR) EQ-VAS score	80.0 (50.0–90.0)	80.0 (70.0–90.0)	80.0 (60.0–90.0)
**Psoriasis-related work productivity loss and activity impairment**			
Patients employed, n (%)	41 (82.0%)	41 (82.0%)	36 (72.0%)
Median (IQR) WPAI-PSO domain scores			
Absenteeism	0.0 (0.0–0.0)	0.0 (0.0–0.0)	0.0 (0.0–0.0)
Presenteeism	0.0 (0.0–0.0)	0.0 (0.0–10.0)	0.0 (0.0–15.0)
Work productivity loss	0.0 (0.0–0.0)	0.0 (0.0–10.0)	0.0 (0.0–18.3)
Activity impairment	0.0 (0.0–0.0)	0.0 (0.0–20.0)	0.0 (0.0–30.0)
**Patient satisfaction with control of psoriasis**			
Percentage of patients with scores on the single-item 7-point Likert-type scale, n (%)			
Satisfied	48 (96.0%)	44 (88.0%)	44 (88.0%)
Completely satisfied	34 (68.0%)	33 (66.0%)	28 (56.0%)
Mostly satisfied	13 (26.0%)	8 (16.0%)	12 (24.0%)
Somewhat satisfied	1 (2.0%)	3 (6.0%)	4 (8.0%)
Uncertain (either satisfied or dissatisfied)	1 (2.0%)	3 (6.0%)	4 (8.0%)
Dissatisfied	1 (2.0%)	3 (6.0%)	2 (4.0%)
Somewhat dissatisfied	0 (0.0%)	1 (2.0%)	0 (0.0%)
Mostly dissatisfied	0 (0.0%)	1 (2.0%)	1 (2.0%)
Completely dissatisfied	1 (2.0%)	1 (2.0%)	1 (2.0%)

N: total number of patients; DLQI: Dermatology Life Quality Index IQR: interquartile range; EQ-5D-5L: EuroQoL 5 Dimensions 5 Levels; n (%): number (percentage) of patients in each category; EQ-VAS, EuroQol Visual Analogue Scale; QoL: quality of life; WPAI-PSO: Work Productivity and Activity Impairment Questionnaire for Psoriasis. Note: WPAI-PSO questionnaires were filled in only by employed participants for the absenteeism, presenteeism, and work productivity loss domain, and by all participants for the activity impairment domain. Among the employed participants, 38 (Estonia), 39 (Latvia), and 36 (Lithuania) answered the questions related to work absenteeism/presenteeism and loss; all participants (50 in each country) answered the questions related to activity impairment.

## Data Availability

AbbVie is committed to responsible data sharing regarding the clinical trials we sponsor. This includes access to anonymized, individual, and trial-level data (analysis data sets), as well as other information (e.g., protocols, clinical study reports, or analysis plans), as long as the trials are not part of an ongoing or planned regulatory submission. This includes requests for clinical trial data for unlicensed products and indications. These clinical trial data can be requested by any qualified researchers who engage in rigorous, independent, scientific research, and will be provided following review and approval of a research proposal, Statistical Analysis Plan, and execution of a Data Sharing Agreement. Data requests can be submitted at any time after approval in the US and Europe and after acceptance of this manuscript for publication. The data will be accessible for 12 months, with possible extensions considered. For more information on the process or to submit a request, visit the following link: https://www.abbvieclinicaltrials.com/hcp/data-sharing/ (accessed on 7 January 2025).

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
