# Peer review of "The Real-World Burden of Moderate-to-Severe Psoriasis in Patients Under Systemic Treatment from Baltic Countries: Data from the CRYSTAL Observational Study"

_medicina, 2025, doi:10.3390/medicina61030397_

Round 1
Reviewer 1 Report
Comments and Suggestions for Authors
1. The article outlines a cross-sectional, multicenter, retrospective study design, which is appropriate for capturing real-world data on psoriasis treatment and outcomes. However, the sample size of 50 patients per country may limit the generalizability of the findings. Future studies could benefit from larger sample sizes to better represent the diverse patient populations and treatment responses across the Baltic countries.
2. The findings indicate that a significant proportion of patients were receiving biological agents, particularly TNF inhibitors. This aligns with current treatment guidelines; however, it's crucial to consider that the efficacy of treatment may vary significantly across different populations and ethno-genetic backgrounds. The article suggests that while many patients achieved low PASI scores, the impact on quality of life indicates that some patients might not be achieving optimal disease control.
3. The study effectively utilized the Dermatology Life Quality Index (DLQI) and EQ-5D-5L measures to assess the quality of life of patients. The reported QoL impairments, specifically in domains such as pain and discomfort, highlight the comprehensive impact of psoriasis beyond skin symptoms. Future research should expand on understanding the contextual factors contributing to these impairments, particularly among different demographic groups within the region.
4. The high levels of patient satisfaction reported (88%–96%) are promising; however, the disconnect between satisfaction and QoL could indicate that patients may accept their current level of treatment due to limited alternatives rather than optimal outcomes. Investigating the reasons behind high satisfaction in the context of ongoing QoL issues could provide deeper insights into patient perspectives and treatment expectations.
5. The study notes a considerable prevalence of comorbid conditions among psoriasis patients, particularly in Latvia and Lithuania. Comorbidities can significantly affect treatment outcomes and patient management strategies, emphasizing the need for a holistic approach to psoriasis care that addresses both skin disease and associated health issues.
6. The article discusses differences in treatment patterns and biologics usage among the three Baltic countries, likely influenced by varying healthcare policies and reimbursement strategies. These findings underscore the importance of understanding regional healthcare systems and guidelines, suggesting that tailored approaches may maximize treatment effectiveness while ensuring equitable access to care across populations.
Author Response
Reviewer #1 comments: |
Response |
Manuscript changes summary |
1. The article outlines a cross-sectional, multicenter, retrospective study design, which is appropriate for capturing real-world data on psoriasis treatment and outcomes. However, the sample size of 50 patients per country may limit the generalizability of the findings. Future studies could benefit from larger sample sizes to better represent the diverse patient populations and treatment responses across the Baltic countries.
|
Thank you for this comment. We acknowledged this limitation and we agree that larger sample sizes would bring additional insights into patient disease and treatment characteristics. Following your comment, we further built into this potential direction that future studies may have. |
Discussion, lines 837-839 To better characterize the disease and treatment response in more diverse patient populations across the Baltic countries, future studies including larger sample sizes are needed. |
2. The findings indicate that a significant proportion of patients were receiving biological agents, particularly TNF inhibitors. This aligns with current treatment guidelines; however, it's crucial to consider that the efficacy of treatment may vary significantly across different populations and ethno-genetic backgrounds. The article suggests that while many patients achieved low PASI scores, the impact on quality of life indicates that some patients might not be achieving optimal disease control.
|
Thank you for this insightful comment. Indeed, at least two-thirds of patients in each country were treated with biological monotherapy, mainly a TNF inhibitor. Also, your comment referring to the impact on quality of life (QoL) is one of the main conclusions in our study: a negative impact on the QoL is still reported, despite low absolute PASI scores and patient satisfaction with their evolution under current treatment. We hence slightly rephrased the final conclusion. |
Conclusions, lines 845-847 Nevertheless, However, a negative impact of the disease on the QoL is still reported, underscoring the need indicating that to further optimize the therapeutic treatment strategies can be further optimized for a greater control of disease.
|
3. The study effectively utilized the Dermatology Life Quality Index (DLQI) and EQ-5D-5Lmeasures to assess the quality of life of patients. The reported QoL impairments, specifically in domains such as pain and discomfort, highlight the comprehensive impact of psoriasis beyond skin symptoms. Future research should expand on understanding the contextual factors contributing to these impairments, particularly among different demographic groups within the region. |
Thank you for this comment. Indeed, the impact of the disease needs to be further explored to better understand the contributors to the impairments described by patients with moderate-to-severe psoriasis. |
Discussion, lines 800-802 An impact on the QoL was observed, especially in terms of pain/discomfort, emphasizing the multifaceted consequences of psoriasis that should be further explored. |
4. The high levels of patient satisfaction reported (88%–96%) are promising; however, the disconnect between satisfaction and QoL could indicate that patients may accept their current level of treatment due to limited alternatives rather than optimal outcomes. Investigating the reasons behind high satisfaction in the context of ongoing QoL issues could provide deeper insights into patient perspectives and treatment expectations.
|
Indeed, you are very right. Exploring to a larger extent the patient perspective, the treatment expectations and how the patients value the treatment objectives become essential for the concordance of treatment objectives and to better understand the disconnect between satisfaction and QoL. |
Discussion, lines 814-816 The satisfaction of patients satisfaction with their current systemic therapy was also high, with 88.0%–96.00% of patients in the three Baltic states being satisfied with their treatment, compared to 90.5% of patients from all Central and Eastern European CEE countries included in the study [24]. Although these are encouraging results, the disease impact on QoL despite current psoriasis treatment should be explored to a larger extent at patient level. |
5. The study notes a considerable prevalence of comorbid conditions among psoriasis patients, particularly in Latvia and Lithuania. Comorbidities can significantly affect treatment outcomes and patient management strategies, emphasizing the need for a holistic approach to psoriasis care that addresses both skin disease and associated health issues.
|
True, the proportion of patients with ongoing concomitant illnesses was highest in Latvia (76%) and Lithuania (54%), and overall ³62% of patients in all countries presented comorbidities. We have added now more details regarding the most prevalent conditions in each country for a full picture, and we made a note regarding the need for a comprehensive patient management. Thank you for highlighting this direction.
|
Results, Table 1 Comorbidities with an incidence of at least 10% in each country have been added, section Comorbidities.
Note under Table 1, Line 528-530 Note: Comorbidities with an incidence of at least 10% in each country have been added. The comorbidities were classified using the Medical Dictionary for Regulatory Activities (MedDRA) v24.0.
Discussion, lines 816-818 In addition, the high proportion of patients with comorbidities across countries (62.0%-78.0%) underlines the importance of a comprehensive patient management, aiming to control all associated diseases. |
6. The article discusses differences in treatment patterns and biologics usage among the three Baltic countries, likely influenced by varying healthcare policies and reimbursement strategies. These findings underscore the importance of understanding regional healthcare systems and guidelines, suggesting that tailored approaches may maximize treatment effectiveness while ensuring equitable access to care across populations. |
Thank you for this valuable comment. Indeed, this was our intention, to bring more recent insights into the treatment patterns and biologics use across the three countries, that could be used as a reference point for future studies.
|
|
Please see the attachment.

Reviewer 2 Report
Comments and Suggestions for Authors
Thank you to the authors for their assessment of moderate-to-severe psoriasis in the Baltic countries. The paper provides valuable insights. The differences between the three countries were well stratified, and overall, it is a well-written manuscript.
Major Comments:
-The title is lengthy and could be shortened for greater impact.
-The section on “ongoing comorbidities” requires further elaboration. What specific comorbidities were identified? These are not detailed in either the main text or the supplemental material.
-For “monotherapy,” it would be helpful to include a breakdown of the therapies used or, at the very least, specify them in Table 3. The same applies to “conventional” agents.
-Clarify how “combination therapy” differs from other two-drug regimens (e.g., methotrexate + etanercept).
-I recommend them to submit the figures as separate high-quality images to improve readability.
Minor Comments:
-Change “all in Latvia” to “100% in Latvia” in the abstract for greater precision.
-The number “4,622,594” is very specific. Consider rounding it to “4.6 million” to better align with the phrase “estimated.”
Author Response
Reviewer #2 comments: |
Response |
Manuscript changes summary |
Major comments |
||
-The title is lengthy and could be shortened for greater impact. |
Indeed, the initial title was lengthier because we wanted to be aligned with the study protocol, but you are right, this could diminish the impact on readers. In this case, our proposal would be to shorten the title to The real-world burden of moderate-to-severe psoriasis in patients under systemic treatment from Baltic countries: data from the observational CRYSTAL study.
|
Title, lines 2-4 Disease severity, treatment patterns and quality of life in patients with The real-world burden of moderate-to-severe psoriasis in patients under receiving systemic treatment in routine clinical settings: data for the from Baltic countries: data from the CRYSTAL observational study |
-The section on “ongoing comorbidities” requires further elaboration. What specific comorbidities were identified? These are not detailed in either the main text or the supplemental material.
|
Following your comment, we added in the Table 1 the overall comorbidities (classified per system, organ and class as per MedDRA v24.0) with an incidence at least 10% in each country. Indeed, this would provide a deeper understanding of the comorbidities burden in each country. An additional comment on the high prevalence of comorbidities has been added in Discussions. |
Results, Table 1 Comorbidities with an incidence of at least 10% in each country have been added, section Comorbidities.
Note under Table 1, Line 528-530 Note: Comorbidities with an incidence of at least 10% in each country have been added. The comorbidities were classified using the Medical Dictionary for Regulatory Activities (MedDRA) v24.0.
Discussion, lines 816-818 In addition, the high proportion of patients with comorbidities across countries (62.0%-78.0%) underlines the importance of a comprehensive patient management, aiming to control all associated diseases. |
- For “monotherapy,” it would be helpful to include a breakdown of the therapies used or, at the very least, specify them in Table 3. The same applies to “conventional” agents.
|
The current systemic treatment has been detailed for both monotherapy (biological/non-biological, and further on, class and active substance, including for conventional treatments) and combination therapy (classes & active substance) in the Table 3, under first section Current systemic treatment for psoriasis. In case the comment refers to prior treatments, this information has been summarized only for biological and non-biological treatments in the main text, the first paragraph under Table 2, lines 539-545. |
- |
-Clarify how “combination therapy” differs from other two-drug regimens (e.g., methotrexate +etanercept). |
Combination therapy (and percentage of patients receiving a specific combination) is detailed in the Table 3, under first section Current systemic treatment for psoriasis. Combination therapy included (i) TNF inhibitors + conventional agents (methotrexate with adalimumab, infliximab or etanercept), in Estonia and Lithuania, but not in Latvia, (ii) IL-12/23 inhibitor + conventional agent (ustekinumab+methotrexate) in Estonia and Lithuania, but not in Latvia. |
- |
-I recommend them to submit the figures as separate high-quality images to improve readability. |
Thank you for this observation. Following your suggestion, we improved the resolution of both figures (Figure 1 and Figure 2), which will be also provided also as separate files. |
Revised figures 1 (line 598) & 2 (line 638) |
Minor comments |
||
-Change “all in Latvia” to “100% in Latvia” in the abstract for greater precision. |
Thank you for this suggestion, we have now amended accordingly the abstract. |
Abstract, lines 28-30 Most patients (78.0% in Estonia, all 100% in Latvia, and 68.0% in Lithuania) were receiving monotherapy with biological agents, mainly TNF inhibitors. |
-The number “4,622,594” is is very specific. Consider rounding it to “4.6 million” to better align with the phrase “estimated.” |
Thank you for this suggestion, we followed it and rounded the exact number for a better alignment with the term “estimated”. |
Introduction, lines 44-45 The incidence of psoriasis shows an increasing trend over time since the 1990s [3] and in 2019, 4,622,594 4.6 million incident cases were estimated to occur globally [4]. |
Please see the attachment.
